# Autophagy and Inflammation: Regulatory Roles in Viral Infections

**DOI:** 10.3390/biom13101454

**Published:** 2023-09-27

**Authors:** Li Chen, Limin Yang, Yingyu Li, Tianrun Liu, Bolun Yang, Lei Liu, Rui Wu

**Affiliations:** 1School of Medicine, Jiamusi University, Jiamusi 154007, China; 218043044@stu.jmsu.edu.cn (L.C.); yli083978@gmail.com (Y.L.); trun9616@gmail.com (T.L.); ybljmsu2023@gmail.com (B.Y.); 2School of Medicine, Dalian University, Dalian 116622, China; yanglimin@dlu.edu.cn

**Keywords:** NLRP3 inflammasome, autophagy, inflammation, viral infection, viral replication

## Abstract

Autophagy is a highly conserved intracellular degradation pathway in eukaryotic organisms, playing an adaptive role in various pathophysiological processes throughout evolution. Inflammation is the immune system’s response to external stimuli and tissue damage. However, persistent inflammatory reactions can lead to a range of inflammatory diseases and cancers. The interaction between autophagy and inflammation is particularly evident during viral infections. As a crucial regulator of inflammation, autophagy can either promote or inhibit the occurrence of inflammatory responses. In turn, inflammation can establish negative feedback loops by modulating autophagy to suppress excessive inflammatory reactions. This interaction is pivotal in the pathogenesis of viral diseases. Therefore, elucidating the regulatory roles of autophagy and inflammation in viral infections will significantly enhance our understanding of the mechanisms underlying related diseases. Furthermore, it will provide new insights and theoretical foundations for disease prevention, treatment, and drug development.

## 1. Introduction

The intricate interplay between autophagy and inflammation during viral infections constitutes a significant area of research. In the course of viral infections, the activation and regulation of autophagy and inflammasomes play pivotal roles in maintaining cellular homeostasis and countering viral invasion [1,2]. Autophagy, a conserved intracellular degradation process, initiates the formation of double-membrane vesicles called autophagosomes. Through this process, autophagy recycles biological macromolecules and organelles by transferring them to lysosomes for degradation. By eliminating intracellular surplus material and damaged organelles, autophagy facilitates the recycling of cellular components, thereby upholding cellular homeostasis under stressful conditions. The core component of the inflammatory response is the inflammasome, which assembles through intracellular pattern recognition receptors to form multiprotein complexes. Inflammasomes drive the occurrence of inflammatory reactions by releasing inflammatory factors. They induce the activation of caspase-1 and facilitate the release of subsequent cytokines, including IL-1β and IL-18, which play a crucial role in resisting pathogenic microbial infections and maintaining immune homeostasis [3]. However, an overactive inflammasome can also trigger a cytokine storm, leading to various inflammatory diseases and exacerbating tissue damage. In viral infections, a complex interplay exists between autophagy and inflammation. Autophagy can inhibit inflammatory responses and mitigate tissue damage caused by excessive inflammasome activation. Conversely, inflammasomes assist in viral clearance and the elimination of harmful substances by inducing autophagosome formation. Therefore, this article aims to review three main aspects: the formation and activation of autophagy and inflammation, the roles of autophagy and inflammation in viral infections, and their mutual regulation.

Viral infection is a complex biological process involving multiple facets of the host immune system. During this process, autophagy and inflammation play crucial roles. Unraveling the interaction between autophagy and inflammation will aid in uncovering viral evasion mechanisms, providing new avenues for vaccine development, and identifying novel drug targets, thereby offering fresh insights and strategies for treating and preventing viral infections. This article aims to comprehensively review three main aspects: the formation and roles of autophagy and inflammation, the involvement of autophagy and inflammation in viral infections, and their intricate interplay. By exploring the intricate relationship between autophagy and inflammation, we aim to uncover novel mechanisms and aspects of viral infections, providing fresh insights and approaches for future treatments and immune modulation in viral infections.

## 2. Autophagy

In 1963, Belgian scientist Christian de Duve made a significant discovery during his research when he observed that after rats were perfused with glucagon a large number of lysosomes capable of degrading intracellular structures appeared in liver cells. Subsequently, he introduced the concept and definition of “autophagy” for the first time at the Lysosome International Conference. In the early 1990s, the research team led by Yoshinori Ohsumi identified the process of autophagy in yeast and identified 15 autophagy-related proteins (Atg) that played a crucial regulatory role in the autophagy process. The autophagy process can be categorized into three types based on the transport of cellular contents to lysosomes: macroautophagy, microautophagy, and chaperone-mediated autophagy [4,5]. Macroautophagy is the most important and relatively conserved form of autophagy. During this process, protein aggregates and misfolded proteins are sequestered within double-membrane vesicles to form autophagosomes, which then undergo degradation through fusion with lysosomes. Microautophagy involves the direct phagocytosis of substances that require degradation through lysosomes [6]. Molecular chaperone-mediated autophagy (CMA) relies on chaperone molecules such as heat shock protein 70 (Hsc70) in the cytoplasm. These chaperones recognize target proteins containing specific motifs and bind to positively charged amino acid residues on lysosome-associated membrane protein 2A (LAMP2A), guiding the targeted proteins to the vicinity of lysosomes. Subsequently, the interaction of LAMP2A with chaperone complexes induces the oligomerization of LAMP2A, thereby promoting the degradation of target proteins from the cytoplasmic matrix to lysosomes, breaking them down into small molecules for cell recycling [7,8]. The formation process of autophagy can be broadly divided into the following stages: initiation of autophagy, elongation, maturation, fusion of autophagosomes with lysosomes, and, ultimately, the degradation and recycling of cell contents (Figure 1).

Initiation Stage of Autophagy: The initiation process of autophagy is typically triggered by various signals, such as starvation, protein aggregation, pathogen invasion, energy or nutrient deficiency, endoplasmic reticulum (ER) stress, and hypoxia, among others. Under normal circumstances, cells maintain a low level of basal autophagy and sufficient energy. The mTORC1 complex highly phosphorylates ATG13, blocking its interaction with ULK1 and FIP200, thereby inactivating it and inhibiting autophagy [9,10,11]. Simultaneously, mTORC1 inhibits the nuclear translocation of the PI3K type III complex and TFEB, the main transcription factor for autolysosome formation [12]. However, under nutrient deficiency or starvation conditions, mTORC1 activity is inhibited, leading to the dephosphorylation and activation of the ULK1 complex, thereby inducing autophagy.

Elongation and Maturation Phases of Autophagy: This phase is driven by two distinct but interconnected ubiquitin-like coupling systems [13]. The initial mechanism involves the Atg12-Atg5-Atg16L system. Initially, the ATG7 protein binds to the ubiquitin-like protein Atg12 and transfers it to the active cysteine site of Atg10 after activation. Subsequently, Atg12 binds with Atg5, forming the Atg12-Atg5 complex. This complex is further associated with Atg16L, creating the Atg12-Atg5-Atg16L complex [14,15]. The subsequent pathway involves the LC3/ATG8-PE conjugation pathway. In mammals, Atg4 catalyzes the cleavage of the carboxyl terminus of the LC3 precursor to activate LC3, generating LC3-I. Following this, LC3-I undergoes ubiquitination by Atg7 and Atg3 and combines with PE. This modification transforms LC3-I into LC3-II, which frequently serves as a significant indicator of autophagosome formation, reflecting the level of autophagy and its correlation with the development and maturation of autophagosomes [16,17,18]. Both of these ubiquitin-like coupling systems are anchored to the autophagosome membrane, facilitating the elongation of the autophagosome by attracting the participation of other membrane structures. This process enables the autophagosome to encapsulate cytoplasmic components that require degradation, ultimately leading to the formation of the autophagosome.

Fusion, Degradation, and Recycling Stages of Autolysosomes: Following autophagosome formation, autophagosomes fuse with lysosomes to form autolysosomes. Within autolysosomes, acid hydrolases are used to degrade the components of autophagosomes and release nutrients. These nutrients can be transported to the cytoplasm for recycling [19,20].

## 3. Autophagy and Viruses

Autophagy can influence or control various stages within the viral life cycle, including virus adsorption and entry, membrane fusion, replication, protein translation, synthesis, and virus release [21]. A bidirectional regulatory relationship exists between autophagy and viruses. On the one hand, autophagy participates in the host’s immune response, exhibiting either an antiviral or proviral function. On the other hand, certain highly adaptable viruses can develop diverse strategies to encode viral proteins, enabling them to evade or inhibit autophagy or exploit autophagy to augment virus replication or facilitate persistent infection (Figure 2).

### 3.1. Virus-Induced Autophagy

#### 3.1.1. Induction of Autophagy through Interaction with Autophagy-Related Proteins

Viral proteins can regulate the autophagy pathway by interacting with vital autophagy-related proteins, thereby promoting the formation of autophagic vesicles and enhancing autophagic activity. Viral proteins can interact with Beclin1 and LC3 to modulate autophagy initiation. Beclin1 is a crucial regulatory protein involved in autophagic vacuole formation. Beclin1 participates in autophagic signal transduction and vesicle formation by forming complexes with other autophagy-related proteins. Many viral proteins possess the ability to bind to Beclin1 and mimic its activity in the autophagy signaling pathway, thus inducing autophagy. For instance, the hepatitis B virus X (HBx) protein can bind to Beclin1, resulting in an upregulation in Beclin1 expression levels and the induction of autophagy in liver cancer cells [22]. Furthermore, the nonstructural protein NS2 of the respiratory syncytial virus (RSV) can interact with Beclin1, resulting in a dual effect. On the one hand, it prevents Beclin1 degradation and enhances cell autophagic activity. On the other hand, NS2 alleviates Beclin1’s anti-autophagic effect, favoring autophagy in cells [23]. Similarly, the 3D protein of enterovirus 71 (EV71) and the P7/NS5A protein of hepatitis C virus (HCV) can also bind to Beclin1, thus promoting the formation of autophagosomes [24,25,26]. LC3 is another crucial protein involved in autophagy, playing a role in the elongation and closure of the autophagosome membrane. Specific viral proteins, like the Gag protein of human immunodeficiency virus (HIV), the M protein of human parainfluenza virus 3 (HPIV3), and the VP1 protein of foot-and-mouth disease virus (FMDV), can interact with LC3 to induce autophagosome formation [27,28,29]. This interaction not only contributes to autophagosome formation but also facilitates viral replication. In addition, infection with Sendai virus (SeV) and vesicular stomatitis virus (VSV) induces the formation of GFP-LC3B dots and promotes the conversion of LC3-I into LC3-II. At the same time, the expression of CCDC50 significantly increases in THP-1 cells. An autophagy receptor can interact with p62 and LC3B to participate in virus-induced autophagy [30]. Beyond Beclin1 and LC3, various other autophagy-related proteins, including Rab5, Vps34, ATG5, and ATG12, can also interact with specific viral proteins to further regulate the autophagy pathway. For instance, the NS4B protein of HCV forms a complex with Rab5 and Vps34, while the NS5B protein interacts with ATG5 and ATG12 to modulate autophagy [31,32].

#### 3.1.2. Modulation of Autophagy Signaling Pathways for Inducing Autophagy

Various factors influence the occurrence of autophagy and can be regulated through distinct signaling pathways. Recently, the PI3K/Akt/mTOR signaling pathway has garnered significant attention within autophagy and virus research. For instance, the capsid protein VP1 of the Seneca Valley virus (SVV) can enhance AKT and AMPK phosphorylation, suppress mTOR activity, and induce autophagy via the AKT-AMPK-mTOR pathway. Additionally, the VP3 and 3C proteins of SVV activate autophagy through the ERK-mTOR and p38 MAPK-mTOR pathways [33]. In the Zika virus (ZIKV) context, the NS4A and NS4B proteins inhibit the Akt-mTOR signaling pathway after infecting human embryonic neural stem cells. This inhibition results in impaired neurogenesis and abnormal autophagy activation, promoting viral replication [34]. Moreover, the small molecule RV-vsRNA1755, encoded by the NSP4 gene of rotavirus (RV), targets and inhibits the expression of IGF1R in host cells. This action subsequently blocks the downstream PI3K/Akt/mTOR signaling pathway, activating cell autophagy and ultimately promoting viral replication [35]. Similar mechanisms are observed in coxsackie virus A16 (CA16), FMDV, and EV71. All these viruses promote replication through the PI3K/Akt/mTOR pathway [36,37,38]. Additionally, coxsackievirus B3 (CVB3) and Epstein–Barr virus (EBV) infections activate the ERK signaling pathway in cells, leading to the induction of autophagy [39,40].

#### 3.1.3. Induction of Autophagy through ER Stress

ER stress constitutes a protective cellular response triggered by the accumulation of misfolded and unfolded proteins within the ER lumen and disruptions in Ca^2+^ balance [41]. Many viruses induce ER stress following infection (Table 1). For example, HCV replication induces autophagy through ER stress, which facilitates the formation of autophagosomes. The abundance of misfolded proteins due to ER stress triggers the UPR. Specific UPR regulators such as PERK, ATF6, IRE1a, and CHOP are implicated in HCV-mediated autophagy [42]. Newcastle disease virus (NDV) replication induces autophagy in human cancer cells, where the expression of its nucleocapsid protein (NP) and phosphoprotein (P) disrupts ER homeostasis, inhibiting the PERK and ATF6 pathways, which activate the UPR response, leading to ER stress and the induction of autophagy [43]. Additionally, a study indicated that the 2C and 3D proteins of EMCV can promote the production of LC3 II and the degradation of p62 protein. These observations suggest that these proteins induce autophagy by activating the ER stress signaling pathway and modulating unfolded protein expression [44].

### 3.2. Antiviral Effect of Autophagy

The initial evidence supporting the role of autophagy in antiviral defense originates from studies on Sindbis virus (SINV) infection. Overexpression of Beclin1 has demonstrated the ability to reduce viral load, mitigate viral pathogenesis, and enhance survival rates following the intracranial injection of SINV. Investigations on mice with neuron-specific *Atg5* knockout have shown that the loss of the autophagy-related protein *Atg5* gene increases the susceptibility of the mouse central nervous system to SINV infection. In vitro research has also shown that the p62 protein directly interacts with viral capsid proteins, promoting the survival of infected cells by aiding in the targeting and removal of viral proteins through autophagy [46].

As an integral part of the immune system, autophagy plays a crucial role in innate immunity and actively contributes to host antiviral defense by influencing antigen presentation to regulate and enhance adaptive immunity [47,48]. The major histocompatibility complex (MHC) is pivotal in shifting from innate to adaptive immunity. Studies have illustrated that autophagy plays a significant role in the MHC I and MHC II antigen presentation process [49]. MHC class I molecules present endogenous antigens that activate CD8(+) T cells, while MHC class II molecules showcase exogenous antigens, activating CD4(+) T cells. Research has substantiated that in mouse macrophages infected with herpes simplex virus type 1 (HSV-1), autophagy enhances MHC-I surface expression via viral glycoprotein B, leading to improved antigen presentation and the activation of CD8(+) T cells [50].

### 3.3. Antiviral Mechanisms of Virus Escape Mediated by Autophagy

Beyond its role in antiviral defense, other viruses can also exploit autophagy. Through intricate interactions with the host organism, viruses have evolved strategies to counteract or manipulate autophagy, allowing them to evade recognition and neutralization by the immune system. These manipulations and evasions of autophagy ensure the viruses’ survival and facilitate replication. The evasion mechanisms employed by these viruses are closely linked to their ability to directly hinder autophagosome formation and obstruct autophagosome–lysosome fusion (Table 2).

#### 3.3.1. Viral Inhibition of Autophagosome Formation

Some viruses can produce specific proteins that impede the process of autophagosome formation, thereby disrupting both the generation and breakdown of autophagosomes. The initiation and nucleation of autophagosomes are primarily governed by alterations in the composition of the PI3K type III complex, consisting of Beclin1, VPS34, VPS15, and ATG14. For instance, the neurotoxin protein ICP34.5 encoded by HSV-1 directly targets Beclin1 to obstruct autophagosome formation and enhance the virus’s neurovirulence [54]. Furthermore, the Us3 protein of HSV-1 can also inhibit autophagy by counteracting ULK1 and Beclin1 [55]. Similarly, it has been reported that human cytomegalovirus (HCMV) can impede the formation and maturation of autophagosomes, relying on the N-terminal domains of viral proteins IRS1 and TRS1 to bind to Beclin1, thereby inhibiting autophagy [66]. Additionally, the virulence factor Bcl-2 encoded by γ-herpesviruses, the viral protein ORF16 of Kaposi’s sarcoma-associated herpesvirus (KSHV), and the M11 protein of murine herpesvirus 68 (MHV-68) all interact with Beclin1 to suppress host autophagy [51,52].

Some viruses can disrupt autophagy by affecting both the Atg12-Atg5-Atg16L system and the LC3-PE binding system. For instance, the FLICE protein vFLIP found in gamma-herpesviruses (HVS), Kaposi’s sarcoma-associated herpesvirus (KSHV), and molluscum contagiosum virus (MCV) can hinder autophagy by interacting with LC3 [53]. Furthermore, during the early stages of foot-and-mouth disease virus (FMDV) infection, autophagy might initially support virus replication; however, the virus can evolve mechanisms to suppress autophagy. This can include cleaving the ATG5-ATG12 complex through the 3C protein and reducing expression levels of LC3II and ATG5-ATG12, thus dampening autophagy, diminishing host antiviral responses, and facilitating virus replication [56].

#### 3.3.2. Viruses Block Autophagosome–Lysosome Fusion

Viruses can subvert autophagy-mediated antiviral activity by obstructing the fusion of autophagosomes with lysosomes. For example, the M2 matrix protein encoded by Influenza A virus (IAV) is involved in multiple stages of virus replication. The M2 protein not only induces the formation of autophagosomes to promote self-replication but also inhibits autophagy by preventing the fusion between autophagosomes and lysosomes [64]. Once formed, autophagosomes combine with lysosomes to create autolysosomes, eliminating surplus cellular components and maintaining cellular metabolism. However, the M2 protein disrupts the fusion of autophagosomes and lysosomes, accumulating many autophagosomes and further facilitating viral replication. HIV also employs a similar strategy. During persistent infection, the HIV Nef protein binds to the autophagy protein Beclin1, resulting in mTOR activation, TFEB phosphorylation, and cytoplasmic sequestration, ultimately inhibiting lysosomal autophagy. This hinders HIV degradation during the early stages of invasion and enhances the generation and release of virions. As a result, it facilitates HIV replication [62,63]. HPIV3 induces incomplete autophagy and provides suitable conditions for viral replication by blocking autophagosome–lysosome fusion. The P protein of HPIV3 binds to SNAP29 and hampers the interaction between SNAP29 and syntaxin17, thus hindering the fusion of autophagosomes and lysosomes mediated by these two host proteins and increasing the production of extracellular virus particles [59]. CA16 infection causes incomplete autophagy, and its 2C and 3C proteins prevent the fusion of autophagosomes and lysosomes [36]. KSHV and EBV viruses also obstruct autophagosome–lysosome fusion by downregulating Rab7 [57,58]. CVB3, FMDV 2C protein, and the SARS-CoV-2 ORF3a protein also inhibit autophagy-mediated antiviral activity by blocking the fusion of autophagosome and lysosome (Table 2) [60,61,65].

## 4. Inflammation

The body’s immune system is divided into innate immunity and adaptive immunity. Innate immunity serves as the initial defense against tissue damage and infections caused by pathogenic microorganisms. It recognizes various stimuli through pattern recognition receptors (PRRs), including exogenous pathogen-associated molecular patterns (PAMPs) and endogenous damage-associated molecular patterns (DAMPs), thereby activating host defense and inflammatory responses [67,68]. Numerous inflammasomes have been identified, such as NLRP3, NLRP6, NLRP7, NLRP12, IPA, and AIM2 [69,70]. These inflammasomes are mainly formed by PRRs, including the NLR, ALR, and Pyrin family [71]. The NLRP3 inflammasome, a crucial member of the NLR protein family, has received significant attention in recent research. It is a highly conserved pattern recognition receptor in cells that responds to endogenous danger signals, playing a vital role in innate immunity and stimulating the body’s immune response. The NLRP3 inflammasome consists of the NLRP3 protein, ASC (PYCARD), and the effector molecule caspase-1. NLRP3 is a trimeric protein comprising an N-terminal pyrin domain (PYD), a central nucleotide-binding oligomerization domain (NACHT), and a leucine-rich C-terminal repeat domain (LRR) [72,73].

The inflammasome activation is a pivotal process in inflammation and is subject to strict regulation. Two main pathways are involved: the classical signaling pathway, dependent on caspase-1, and the non-canonical signaling pathway, dependent on caspase-4/5/11 (Figure 3). The classical signaling pathway, depending on caspase-1, has been extensively studied and involves two steps [71,74]. First, the initiation phase of the inflammasome requires NLRP3 to recognize DAMPs and PAMPs, which, in turn, mediate the expression of cytokines such as IL-1β and TNF-α. This process activates the NF-κB signaling pathway and promotes the expression of NLRP3, IL-1β, and IL-18 precursor proteins. The second phase is the activation phase of the inflammasome. Following initiation, NLRP3 is activated in response to bacterial, fungal, and viral infections and DAMPs-mediated sterile inflammation [75]. The NLRP3 inflammasome can be activated through various mechanisms, and several common activation pathways have been identified, including the ion activation pathway, mitochondrial damage, generation of reactive oxygen species, and lysosome rupture. The ion activation pathway involves NLRP3 activators altering the ion concentration of K^+^, Ca^2+^, and Cl^−^ inside and outside the cell. K^+^ outflow is a common mechanism for inflammasome activation. The P2X7R cation channel functions as a K^+^ efflux channel. Upon the release of a high concentration of ATP, the P2X7R purinergic receptor on the cell membrane is activated, leading to the induction of K^+^ efflux. Simultaneously, the recruitment of the Pannexin-1 channel protein results in the opening of the corresponding channels, thus activating the NLRP3 inflammasome [76]. Reactive oxygen species (ROS) act as upstream factors for NLRP3 inflammasome assembly, and mitochondrial damage releases both mitochondrial DNA (mtDNA) and mitochondrial ROS (mtROS), which are implicated in the activation of inflammasomes [77,78,79]. Lysosomal rupture can trigger the activation of the NLRP3 inflammasome. Crystal substances, including cholesterol crystals and alum crystals, enter the cell through phagocytosis, resulting in lysosomal swelling and rupture. This process leads to the release of cathepsin B, which subsequently facilitates the activation of the NLRP3 inflammasome [80].

Non-canonical signaling pathways dependent on caspase-11/4/5 have been identified. In this pathway, caspase-4 and caspase-5 serve as the mouse and human homologs of caspase-11, respectively. Lipopolysaccharide (LPS) acts as an activator for non-classical inflammasomes. These inflammasomes can bind to caspase-11/4/5, resulting in the direct cleavage of GSDMD and the generation of N-GSDMD. This process mediates the onset of pyroptosis [81,82,83,84,85]. Furthermore, under the stimulation of LPS, caspase-11 can also cleave the Pannexin-1 channel protein, leading to the release of ATP, promoting K^+^ efflux, activating the NLRP3 inflammasome, and subsequently activating caspase-1, which in turn stimulates the secretion of IL-1β and IL-18 [86]. These findings shed light on the intricate regulatory mechanisms involved in the non-canonical inflammasome signaling mediated by caspase-11/4/5 and its implications in cellular responses to bacterial infections.

## 5. Inflammation and Viruses

Once a virus invades a host cell, the corresponding inflammatory signal recognition receptors recognize its viral RNA, proteins, and particles. This recognition triggers the activation of the NLRP3 inflammasome, subsequently releasing factors such as IL-1β and IL-18. Proper inflammasome activation helps the body deal with external metabolic stress. However, it is essential to note that excessive inflammasome activation can also lead to pathological damage. Therefore, activating the NLRP3 inflammasome and its associated inflammatory response represent a double-edged sword for the host in its resistance against viral infections. The NLRP3 inflammasome was first reported to be activated by IAV and SeV. Subsequent studies have indicated that HCV, Adenovirus (AdV), encephalomyocarditis virus (EMCV), RSV, and EV71 can also activate the NLRP3 inflammasome. In recent years, research has identified certain viruses that have developed virulence factors that aim at inhibiting the assembly and activation of the NLRP3 inflammasome. Some examples of these factors include the V protein of measles virus (MV) and SeV and EV71 2A and 3C proteins. These virulence factors prevent excessive inflammatory responses, thus reducing tissue damage and promoting viral replication [2].

### 5.1. Viruses Promote the Activation of the NLRP3 Inflammasome

Viruses can influence the activation of the NLRP3 inflammasome through various mechanisms (Table 3). Viral infections can induce an imbalance in intracellular ion concentrations, leading to mitochondrial damage and lysosome rupture, activating the NLRP3 inflammasome. Furthermore, virus particles or their components can directly interact with the NLRP3 protein within cells, thereby initiating the assembly and activation of inflammasomes (Figure 4).

#### 5.1.1. Ion Concentration Imbalance Mediates the Activation of the NLRP3 Inflammasome

Viral infections can lead to an imbalance in intracellular ion concentrations, leading to an abnormal accumulation or loss of K^+^, Ca^2+^, and Na^+^ ions, activating the NLRP3 inflammasome. This disruption of ion concentration within the cell alters the intracellular environment, thus facilitating the assembly and activation of the inflammasome.

Activation of the NLRP3 inflammasome through K^+^ efflux following viral infection: Under normal circumstances, the concentration of potassium ions inside and outside the cells remains balanced. However, during cellular injury or viral infection, the cell membrane’s potassium ion channels can open, leading to a significant intracellular efflux of K^+^. Recent studies have identified several viruses, including FMDV, EMCV, and VSV, capable of activating the NLRP3 inflammasome by promoting potassium ion efflux. Consequently, this process triggers pyroptosis and the release of inflammatory mediators, such as IL-1β, thereby eliciting neuroinflammatory responses. In the case of FMDV infection, its 2B protein can activate the NLRP3 inflammasome and promote IL-1β secretion through ion channels. Notably, recent studies have indicated that the activation of NLRP3 signaling in FMDV infection is associated with K^+^ efflux and Ca^2+^ influx but independent of mtROS and lysosomal cathepsin B [93,94]. The replication of EMCV and VSV induces lytic cell death, leading to K^+^ efflux, subsequent activation of the NLRP3 inflammasome, and the release of IL-1β, resulting in pyroptosis and necrosis [95,111]. Additionally, proteins such as SARS-CoV 3a protein [91], Mayaro virus (MAYV), HIV-1 gp120 protein, and IAV M2 protein have also been found to activate the NLRP3 inflammasome by promoting K^+^ efflux [87,88,89,90].

Activation of the NLRP3 inflammasome occurs following viral infections by mobilizing Ca^2+^: The core protein of the HCV is capable of mobilizing intracellular Ca^2+^ through a mechanism associated with phospholipase-C activation. This activity consequently promotes the activation of the NLRP3 inflammasome [112]. Similarly, the FMDV 2B protein localizes to the ER, disrupting membrane integrity and altering the Ca^2+^ concentration in the host cell, thereby triggering the activation of the NLRP3 inflammasome [93,94]. Moreover, some viral envelope proteins, such as the E protein of SARS-CoV, the porin 2B protein of EMCV, and human rhinovirus (HRV), are found in the Golgi apparatus. These proteins reduce the intracellular Ca^2+^ concentration by forming Ca^2+^ channels, activating the NLRP3 inflammasome, and influencing IL-1β secretion [92,96,97].

Moreover, recent studies have revealed that some viruses have channel proteins in the Golgi apparatus and other cytoplasmic structures. These proteins can interact with the cell membrane to create selective ion channels to facilitate the transport of specific ions, ultimately activating the inflammasome. For instance, the M2 protein of IAV is localized to the Golgi apparatus, where it acidifies the Golgi compartment. As an H^+^-selective ion channel on the Golgi membrane, the M2 protein induces H^+^ efflux when the Golgi lumen becomes acidic. This, in turn, triggers the activation of the plasma membrane channel and activates the NLRP3 inflammasome [88,89]. Another example involves the hydrophobin SH encoded by RSV. This protein can accumulate in the lipid raft structure of the Golgi apparatus and form selective ion channels. Consequently, it induces the translocation of NLRP3 from the cytoplasm to the Golgi apparatus, activating the inflammasome. Remarkably, inhibition of ion channel activity and lipid raft dissociation effectively blocks inflammasome activation during the treatment of infected cells, underscoring the pivotal role of the lipid raft structure formed by the SH protein in the Golgi apparatus during inflammasome activation [99].

#### 5.1.2. Mitochondrial Damage Activates the NLRP3 Inflammasome

Viral infections may also lead to mitochondrial damage. Such damage often results in the release of molecular components from the mitochondria into the cytoplasm, exhibiting hallmarks of damage and pro-inflammatory properties. Although the precise mechanism through which mitochondrial damage activates the NLRP3 inflammasome remains incompletely understood, several studies have suggested the potential involvement of various processes. These processes include the release of mtDNA, generation of ROS, alteration of ion channels, and activation of specific signaling pathways. ROS plays a pivotal role as a regulator of NLRP3 inflammasome activation induced by viral infections. For instance, dengue virus (DENV) NS2A and NS2B proteins have been shown to promote the assembly of NLRP3 inflammasome complexes, leading to caspase-1 activation and subsequent IL-1β secretion [101]. Additionally, HBx localizes to mitochondria and enhances NLRP3 inflammasome-mediated inflammation and pyroptosis by upregulating mtROS production [102]. The RIP1-RIP3-DRP1 pathway is a common route for activating the NLRP3 inflammasome during viral infections. For instance, VSV infection enhances the interaction between RIP1 and RIP3, activating DRP1 and causing aberrant mitochondrial fission and damage. Consequently, this process ultimately facilitates the activation of the NLRP3 inflammasome [103]. Furthermore, NLRP3 has the capacity to interact with Mitofusin-2 in the context of viral infections. Following EMCV infection, NLRP3 is implicated in facilitating mitochondrial membrane potential generation. This, in turn, leads to the translocation of NLRP3 to the mitochondria, where it forms a complex with Mitofusin-2, subsequently enhancing the activation of the NLRP3 inflammasome [104].

#### 5.1.3. Lysosomal Rupture Activates the NLRP3 Inflammasome

Lysosomal disruption leads to the release of cathepsin B from lysosomal contents, which stimulates the activation of the NLRP3 inflammasome. It has been reported that the structural protein NA of IAV, mainly NA of the H5N1 virus, may bind to lysosome-associated membrane proteins 1 and 2, thereby promoting the deglycosylation of LAMPs and inducing lysosomal endosome rupture, resulting in the release of cathepsins into the cytoplasm. This process ultimately leads to an increase in viral load and enhances cell death [105]. Moreover, AdV infection can also destroy the lysosomal membrane, leading to the release of cathepsin B and subsequent activation of the NLRP3 inflammasome. This observation suggests that different viral infections may exploit lysosomal disruption and the release of cathepsins to trigger inflammatory responses and NLRP3 inflammasome formation [106].

#### 5.1.4. Affecting Inflammasome Assembly and Activating the NLRP3 Inflammasome

Some viral proteins can activate the NLRP3 inflammasome by impacting its assembly. For instance, the N protein in SARS-CoV-2 interacts with NLRP3, aiding the connection between NLRP3 and ASC. This interaction ultimately promotes the formation of the inflammasome [107]. Similarly, the 3D protein of EV71 engages with the NLRP3 protein, forming a 3D-NLRP3-ASC ring structure. This interaction substantially impacts the assembly and activation of the inflammasome complex, ultimately resulting in the activation of IL-1β [108]. Furthermore, the NS5 protein of ZIKV binds to the LRR and NACHT domains of NLRP3 in the cytoplasm to form the NS5–NLRP3–ASC ring domain, which effectively promotes the assembly of the NLRP3 inflammasome complex [109]. These insightful studies shed light on how viral proteins directly or indirectly regulate inflammasome activation through interactions with NLRP3. This enhanced understanding of inflammatory response regulation broadens our knowledge and opens new avenues for researching treatments related to inflammatory diseases associated with viral infections.

### 5.2. Viral Inhibition of NLRP3 Inflammasome Activation

Viral infections activate the NLRP3 inflammasome and inhibit its assembly and activation through interactions (Table 4). For instance, the V proteins of MV, SeV, and Nipah virus, as well as the NS1 protein of IAV, all impede the assembly of the NLRP3 inflammasome. The V protein of the MV engages with NLRP3 to hinder the activation of the NLRP3 inflammasome and the subsequent secretion of IL-1β [113,114]. Although SeV can activate the NLRP3 inflammasome, its structural V protein inhibits the release of IL-1β by preventing NLRP3 inflammasome assembly. Komatsu T [115] found that upon infecting THP1 macrophages with a Sendai virus in which the *V* gene was knocked out, the expression level of IL-1β was significantly increased. The nonstructural proteins NS1 and PB1-F2 of the IAV have inhibitory effects on the NLRP3 inflammasome. Specifically, the NS1 protein significantly suppresses the secretion of IL-18 and IL-1β from THP-1 cells treated with ATP and LPS. It accomplishes this by targeting two signals of the NLRP3 inflammasome, NF-κB in Signal 1 and NF-κB in Signal 2 of NLRP3, inhibiting the inflammasome pathway [116]. In the same vein, the PB1-F2 protein binds to the PYD and LRR domains of NLRP3, inducing a state of autoinhibition in NLRP3. This inhibits the binding of NEK7 and NLRP3, thereby impeding inflammasome assembly and promoting virus escape [117]. Moreover, EBV, AdV, ZIKV NS3, and other viral components have also been identified as inhibitors of NLRP3 inflammasome assembly and activation [118,119,120].

Some viral proteins can inhibit the activation of the NLRP3 inflammasome by mediating the degradation of the NLRP3 protein (Table 4). The EV71 virus, a member of the Picornaviridae Enterovirus genus, is a single-stranded positive-strand RNA virus known for its strong infectivity and pathogenicity. EV71 and the NLRP3 inflammasome exhibit a regulatory relationship wherein they not only promote the activation of the inflammasome but also inhibit its activation through cleavage processes. Research has revealed that the 3C and 2A proteases of EV71 can cleave the NLRP3 protein at the junction of Q225-G226 or G493-L494, thereby hindering the activation of inflammasomes and evading immune system surveillance [121]. Moreover, the HPIV3 C protein can interact with the NLRP3 protein, facilitating its proteasomal degradation and consequently blocking the inflammasome activation [122].

### 5.3. The Role of Cytokines in the Antiviral Immune Process

After a viral infection, the human immune system responds rapidly, producing numerous cytokines that directly participate in antiviral immunity. These cytokines include interferons (IFN), interleukins (IL), and tumor necrosis factor (TNF), among others. They possess direct antiviral effects, inhibiting viral replication and spread, thereby aiding in the clearance of infections. As integral components of adaptive immunity, CD4(+) T cells assist other immune cells by releasing various cytokines, earning them the designation of helper T cells. Depending on the cytokines they produce, CD4(+) T cells can be categorized into two subpopulations: type I helper T cells (Th1) and type II helper T cells (Th2). Th1 cells secrete IFN-γ, IL-2, IL-12, IL-18, TNF-α, and other cytokines, participating in cellular immune responses to organ-specific autoimmune diseases, and are pivotal in combating viral infections. Th2 cells, on the other hand, release IL-4, IL-5, IL-6, IL-9, IL-10, IL-13, and other cytokines, contributing to antibody production and eosinophilic responses [123]. They also play a role in controlling specific viral infections and maintaining the balance between humoral and cellular immunity in the body. In the case of HCMV infection, IL-2 produced by CD4(+) T cells can initiate and enhance nonspecific immunity by activating NK cells in peripheral blood, thereby promoting the clearance of the HCMV virus and inhibiting its activation [124]. Additionally, it can stimulate the phenotypic and functional differentiation of NK cells, leading to increased production of IFN-γ and further encouraging the release of cytokines such as IL-12 and IL-18. This influences the effectiveness of various vaccines, indirectly confirming that HCMV can induce nonspecific immune responses and promote the release of IFN-γ by NK cells, thus exerting an antiviral effect [125]. Upon infection with HRSV, the host protease activates the F protein, resulting in the fusion of the viral shell and the host cell membrane. This facilitates virus entry into the cell, triggering a Th1-type inflammatory response and the production of IFN-γ, TNF-α, and IL-2. These cytokines activate cytotoxic T cells and natural killer cells, promoting the effective clearance of viruses. Conversely, the G protein enables RSV to adhere to host cells, inciting Th2-type inflammatory responses [126,127]. 

Furthermore, under the body’s normal immune state, Th1 and Th2 cytokines maintain a specific ratio, sustaining a dynamic equilibrium. However, the balance between Th1 and Th2 cytokines may shift when the body’s immune status faces challenges. In most cases, a dominant Th1 response aids in virus clearance and recovery. Conversely, if a dominant Th2 response prevails, it may exacerbate the body’s condition. For instance, HRSV infection can disrupt the balance between Th1 and Th2 cellular immune responses, with a predominant Th2 response leading to ineffective virus clearance and an increased risk of pathological damage [128].

## 6. Interaction between Autophagy and the NLRP3 Inflammasome

A complex regulatory relationship exists between autophagy and inflammasomes, with functional interconnections. Autophagy can either stimulate or suppress inflammatory responses, making it a critical regulator of inflammasomes. Conversely, the inflammatory response triggered by inflammasome activation can positively or negatively influence autophagy. The NLRP3 inflammasome can modulate autophagy to maintain a balance in the desired inflammatory response and restrain excessive and detrimental inflammation (Figure 5). This dynamic interplay is essential for maintaining tissue homeostasis [129,130,131].

### 6.1. Autophagy Inhibits NLRP3 Inflammasome Activation

Autophagy suppresses the activation of the NLRP3 inflammasome by regulating autophagosome formation and degradation properties. In 2008, Saitoh T [132] first reported that autophagy can exert an immunosuppressive effect by limiting the activation of inflammasomes. Under normal circumstances, ATG16L1 covalently binds to the ATG12-ATG5 complex to form a multimeric complex, which then binds to LC3-PE to form an autophagosome. The lack of ATG16L1 disrupts the aggregation of the ATG12-ATG5 complex, resulting in the inability of LC3 to bind to PE, affecting the formation of autophagosomes and the degradation of long-lived proteins. Upon LPS stimulation, the content of inflammatory factors, such as caspase-1 and IL-18, in mouse fetal liver macrophages lacking the ATG16L1 gene increased, and IL-1β secretion in macrophages lacking ATG7 or treated with 3-MA increased. Subsequent evidence has further suggested that autophagy can negatively regulate inflammasome activation.

Autophagy prevents NLRP3 inflammasome activation by eliminating endogenous inflammasome activators, including mtROS, released due to mitochondrial damage. For example, during strenuous exercise, mitochondrial stress increases ROS production, triggering an inflammatory reaction within the rat myocardium via NLRP3 inflammasome activation. However, as mtROS accumulates, the process of mitophagy is triggered. This leads to the inhibition of NLRP3 inflammasome activation, alleviating myocardial oxidative stress and the inflammatory response. Consequently, this reduction in myocardial injury occurs [133].

Autophagy can also directly target inflammasome components, such as NLRP3, caspase-1, and ASC, to inhibit the assembly and activation of the NLRP3 inflammasome by degrading them [134,135]. For example, CCDC50 is an autophagy receptor that inhibits inflammatory responses by obstructing the assembly of NLRP3 inflammasomes. CCDC50 can target and degrade K63-ubiquitinated NLRP3, thereby inhibiting the aggregation of NLRP3 and ASC and preventing the formation of ASC spots. This reduction in IL-1β release consequently impedes inflammasome activity [136]. Ubiquitin-specific enzyme 22 (USP22) acts as a negative regulator in the assembly and activation of the NLRP3 inflammasome, promoting ATG5-mediated macroautophagy/autophagy to facilitate NLRP3 degradation [137]. Studies have also found that autophagosomes can sequester pro-IL-1β and caspase-1, thus inhibiting IL-1β secretion and further regulating the inflammatory process [138].

### 6.2. Autophagy Promotes NLRP3 Inflammasome Activation

Autophagy generally functions to inhibit the activation of the NLRP3 inflammasome. However, there are instances where autophagy can also promote NLRP3 inflammasome activation. Dupont et al. have discovered that autophagy can activate NLRP3 inflammasomes. In primary mouse bone marrow-derived macrophages subjected to starvation, autophagy facilitates caspase-1 activation through an Atg5-dependent non-canonical pathway, thereby promoting inflammasome activation and subsequent synthesis and secretion of IL-1β and IL-18 [139]. In addition, studies in a uric acid-induced hyperuricemia model have shown that hyperuricemia upregulates autophagy mediated by the P53 pathway, which induces the activation of the NLRP3 inflammasome. Mechanistically, during uric acid-induced autophagy, the fused autolysosome releases proteinase B, activating the NLRP3 inflammasome. However, treatment with the autophagy inhibitor 3-MA can restrain the autophagy-inflammasome pathway, ameliorating uric acid-induced pyroptosis and preventing the onset of hyperuricemic nephropathy [140].

### 6.3. NLRP3 Inflammasome Regulation of Autophagy

The NLRP3 inflammasome is involved in the activation of autophagy. Studies have been conducted that demonstrate that the NLRP3 inflammasome becomes activated upon macrophage infection by Pseudomonas aeruginosa [141]. In THP-1 macrophages, the overexpression of NLRP3, ASC, or caspase-1 significantly enhances autophagy induced by Pseudomonas aeruginosa infection. Conversely, autophagy is attenuated when these core molecules within the NLRP3 inflammasome are either knocked down or inhibited, thus providing compelling evidence of the NLRP3 inflammasome’s positive regulatory effect on macrophage autophagy.

The NLRP3 inflammasome inhibits autophagy, mainly contributing to neuroinflammation. For instance, Alzheimer’s disease (AD) is a prevalent progressive neurodegenerative disorder characterized by extensive β-amyloid (Aβ) deposits in the brain. These extracellular Aβ deposits activate the NLRP3 inflammasome and trigger the secretion of IL-1β in microglia [142,143]. Autophagy can remove excess Aβ in AD patients. However, the glial maturation factor (GMF) upregulates neuroinflammation in AD patients, inhibiting the removal of Aβ by autophagy [144]. Under normal conditions, NLRP3 inflammasome-mediated autophagy inhibition involves caspase-1 activation, leading to TRIF cleavage. The reduction in the signaling molecule TRIF subsequently affects certain autophagic pathways through the TLR4-TRIF signaling pathway [145]. Furthermore, the knockdown of lincRNA-Cox2 in neuro macrophages and microglia led to a downregulation in NLRP3 and ASC mRNA and protein levels. This knockdown also inhibited inflammasome activation and mitigated the TIR domain-induced cleavage of interferon-β adapter proteins, thereby promoting TRIF-mediated ATG5-dependent autophagy [146].

## 7. Interaction of Autophagy and Inflammation in Viral Infection

A complex interplay exists between autophagy and inflammation during viral infections. Both autophagy and inflammation serve as vital mechanisms through which cells combat pathogens, playing pivotal regulatory roles in the viral infection process. During viral infections, host cells can control inflammasome activation using endogenous regulatory pathways, including autophagy, to prevent undue inflammation that could harm the body. Inflammation and cardiopulmonary injury are primary factors contributing to the morbidity and mortality of COVID-19 patients. Within this context, impaired mitophagy and the subsequent overproduction of ROS are instrumental in inflammation and cardiorespiratory dysfunction. The spike protein S of the SARS-CoV-2 virus connects with the ACE2 receptor on the host cell surface through its receptor-binding domain (RBD) in the S1 subunit, orchestrating the virus’s entry into the host cell. Recent research has demonstrated that following an infection, the spike protein can stimulate the NLRP3 inflammasome by suppressing mitophagy and elevating mtROS production. This promotes the maturation of IL-18 and results in cardiac injury. Nevertheless, enhancing mitophagy effectively reduces spike protein-induced IL-18 expression. In addition, IL-18 inhibition reduces spike-mediated pNF-κB and EC permeability [147,148,149].

In HBV infections, the covalently closed circular DNA (cccDNA) is pivotal in driving viral replication and liver damage, significantly influencing liver fibrosis. Studies suggest that the activation of the STING pathway by the interferon gene stimulator can effectively regulate the autophagy pathway. This regulation subsequently leads to a reduction in the activation of inflammasomes, thus mitigating the extent of liver injury and fibrosis induced by HBV. The signaling activation of STING also exerts a notable inhibitory effect on cccDNA transcription through epigenetic mechanisms. Furthermore, it contributes to the attenuation of HBV-induced liver fibrosis by suppressing macrophage inflammasome activation and promoting the activation of autophagic flux [150].

The exacerbation of Leishmania RNA virus (LRV)-mediated disease hinges on TLR3 activation. This induces the secretion of inflammatory cytokines, such as TNF-α and IL-12, and triggers type I interference via TRIF-mediated IFN production. Previous research has affirmed that type I IFN can induce autophagy [151]. Newer studies suggest that post-LRV infection, TLR3-driven inflammasome-independent cytokine production can incite autophagy, leading to ATG5-mediated degradation of NLRP3 and ASC. This inhibits NLRP3 assembly in macrophages and curtails NLRP3 inflammasome activation, eventually promoting parasite survival [152].

The ssRNA40 fragment of HIV activates the NLRP3 inflammasome in both microglia and astrocytes. This activation triggers the release of inflammatory factors and stimulates the production of TNF-α, IL-1α, and C1q. Concurrently, HIV ssRNA40 encourages the accumulation of damaged mitochondria and reduces autophagy. The induction of autophagy curtails ssRNA40-mediated inflammasome activity and prevents cell death. However, inhibiting autophagy or mitophagy in the context of regulating NLRP3 inflammasome activity results in the excessive release of inflammatory cytokines, the activation of caspase-1, and, ultimately, microglial cell death [153].

In conclusion, the relationship between autophagy and inflammation during viral infections is of paramount significance. Autophagy and inflammation, acting as cellular defense mechanisms against pathogens, are integral in governing viral infections. Yet, given the diverse nature of various viruses, these regulatory mechanisms can vary. Understanding the autophagy–inflammation interplay in viral infections will illuminate the disease’s pathogenesis and offer a theoretical foundation for crafting new therapeutic strategies.

## 8. Conclusions

In recent decades, numerous studies have underscored the pivotal roles that autophagy and inflammation play in maintaining cellular homeostasis and combating viral infections. They can either trigger or suppress the progression of diseases through diverse mechanisms. This review delves deeply into the regulatory mechanisms governing autophagy and inflammation during viral infections, aiming to enhance our comprehension of viral pathogenesis. Nonetheless, several intricacies within this field remain to be elucidated. Firstly, the regulation of autophagy and inflammation varies across different cell types, stimulus conditions, and disease states. Consequently, we still lack precise insights into their mechanisms during viral infections. This variability may stem from genomic, environmental, and host factors, necessitating further research on how these variables influence autophagy and inflammation regulation. Secondly, additional investigations are essential to confirm the potential therapeutic value of autophagy and inflammation. While these processes seem promising as targets for disease treatment, practical challenges persist, including the need for precise intervention and the minimization of potential adverse effects. In forthcoming research, we can further explore the regulatory mechanisms, validate their therapeutic potential, and devise innovative treatment strategies. These endeavors will enhance our comprehension of viral pathogenesis, offer novel insights and strategies for managing related diseases, and significantly contribute to advancing human health.

## Figures and Tables

**Figure 1 biomolecules-13-01454-f001:**
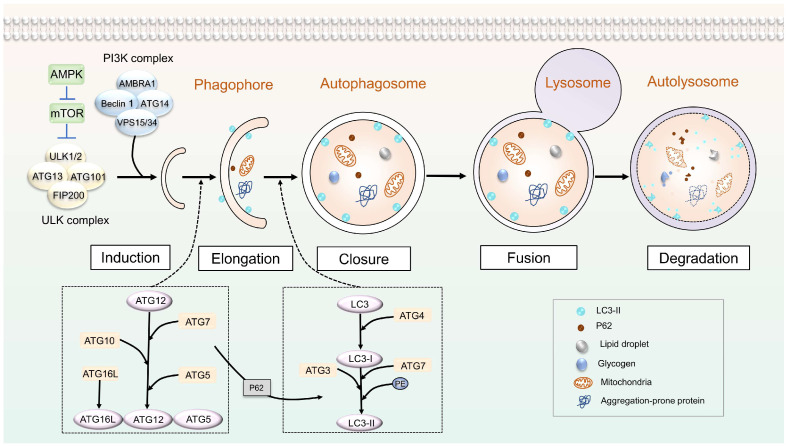
Molecular Mechanism of Autophagy: The formation process of autophagy can be broadly categorized into the following stages: initiation of autophagy, elongation, maturation, fusion of autophagosomes and lysosomes, and, ultimately, degradation and recycling of cell contents. Under starvation conditions, the ULK1 complex is activated, leading to the phosphorylation of ULK1 and Beclin1. Phosphorylated Beclin1, in turn, activates VPS34, forming a PI3K type III complex by associating with VPS15, Beclin1, and ATG14. This complex plays a critical role in the initiation of autophagy. The autophagy process is further facilitated by two integral coupling mechanisms: the Atg12-Atg5-Atg16L system and the LC3/ATG8-PE binding system. These systems are responsible for tagging specific proteins as targets for autophagic degradation, actively participating in the process.

**Figure 2 biomolecules-13-01454-f002:**
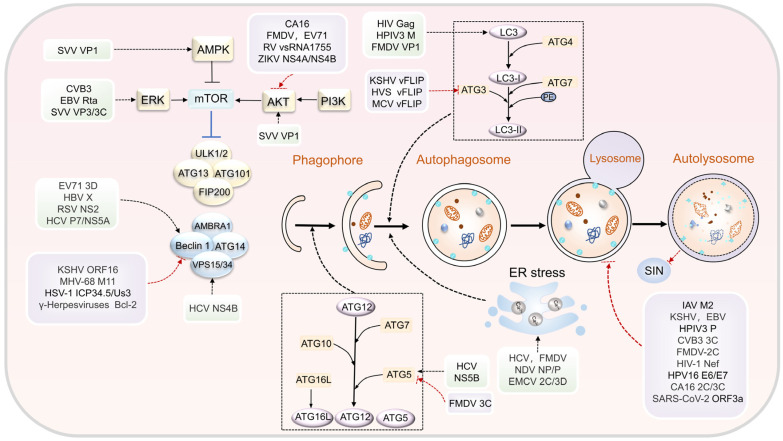
Mechanism of autophagy in viral infection. (1) Induction of autophagy through interaction with autophagy proteins: EV71 3D, HBVX, RSV NS2, and HCV P7/NS5A proteins bind to Beclin1. Similarly, HIV Gag, HPIV3 M, and FMDV VP1 proteins interact with LC3. HCV NS4B forms a complex with Rab5 and Vps34, while NS5B interacts with ATG5 and ATG12 to initiate autophagy. (2) Modulation of autophagy signaling pathways for inducing autophagy: CA16, FMDV, EV71, SARS-CoV-2, RV, ZIKV, and other viruses activate autophagy by inhibiting the PI3K/AKT/mTOR pathway. VP3/3C EBV, CVB3, and SVV proteins activate autophagy through the ERK pathway. SVV VP1 protein induces autophagy through the AKT-AMPK-mTOR pathway. (3) Induction of autophagy through ER stress: HCV, FMDV, NDV, and EMCV trigger cell autophagy by inducing ER stress. (4) Viral inhibition of autophagosome formation: KSHV, MHV-68, HSV-1, and γ-herpesviruses inhibit host autophagy by interacting with Beclin1. The vFLIP proteins of KSHV, MCV, and HVS inhibit autophagy by preventing Atg3 from binding and processing LC3. FMDV 3C protein cleaves the ATG5-ATG12 complex to suppress autophagy and promote virus replication. (5) Viruses block autophagosome–lysosome fusion: KSHV, EBV, HPIV3, and CVB3 prevent the fusion of autophagosomes with lysosomes, a strategy that favors viral replication.

**Figure 3 biomolecules-13-01454-f003:**
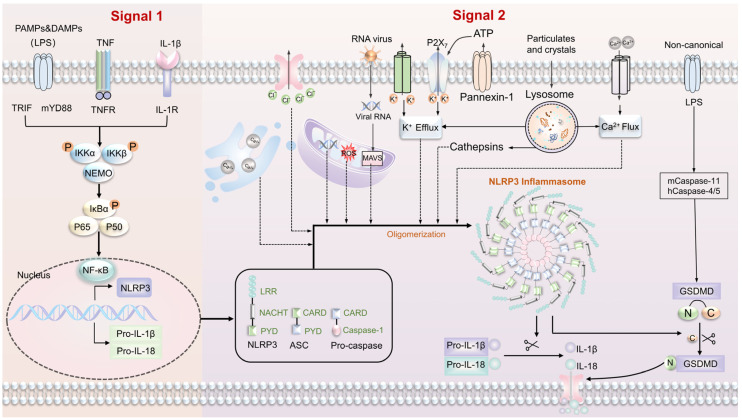
Activation mechanism of the NLRP3 inflammasome. Initiation signal (Signal 1): NLRP3 recognizes DAMPs and PAMPs, activating the NF-κB signaling pathway through mediating cytokines such as IL-1β and TNF-α. In turn, it promotes the activation of NLRP3, Pro-IL-18, and Pro-IL-1β, resulting in the expression of precursor proteins. Activation signal (Signal 2): The NLRP3 inflammasome is triggered through multiple pathways, including the ion activation pathway, mitochondrial damage with reactive oxygen species generation, and lysosome rupture. The NLRP3 inflammasome consists of three main components: the NLRP3 protein, ASC, and the effector molecule caspase-1. Upon activation, caspase-1 cleaves the precursors of IL-1β and IL-18, converting them into their active forms. Subsequently, IL-18 initiates subsequent signaling pathways, participating in various inflammatory reactions and exerting biological effects. Furthermore, NLRP3 can also stimulate caspase-1 to cleave GSDMD, forming N-GSDMD and triggering pyroptosis.

**Figure 4 biomolecules-13-01454-f004:**
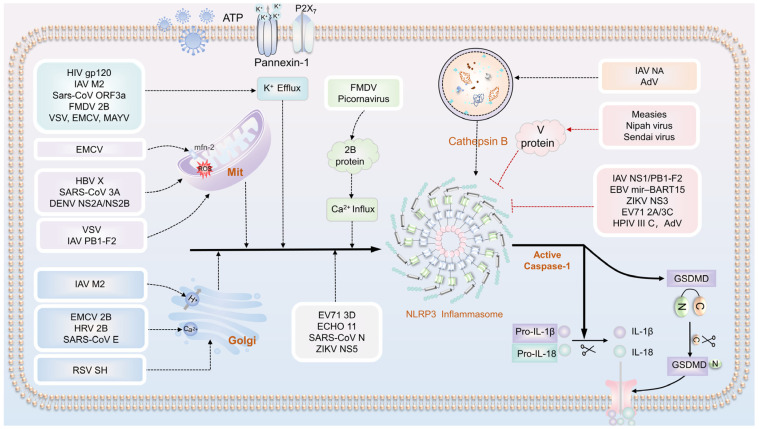
Viruses Promote or Inhibit NLRP3 Inflammasome Activation through Various Mechanisms. (1) Ion concentration imbalance mediates the activation of the NLRP3 inflammasome: HIV gp120, IAV M2, SARS-CoV ORF3a, FMDV 2B, VSV, EMCV, and MAYV induce NLRP3 inflammasome activation through K^+^ efflux. EMCV 2B, HRV 2B, SARS-CoV E, FMDV 2B, and Picornavirus activate the inflammasome by mobilizing Ca^2+^. RSV SH aggregates in the lipid raft structure of the Golgi apparatus, forming selective ion channels that translocate NLRP3 from the cytoplasm to the Golgi apparatus, subsequently leading to inflammasome activation. The IAV M2 protein localizes to the Golgi apparatus and induces H^+^ efflux, further activating the NLRP3 inflammasome. (2) Activation of NLRP3 inflammasome through mitochondrial damage: HBV X, SARS-CoV 3A, and DENV NS2A/NS2B enhance NLRP3 inflammasome-mediated inflammation and pyroptosis by promoting the production of mtROS. VSV and IAV PB1-F2 accelerate mitochondrial damage to activate the NLRP3 inflammasome. EMCV activates the NLRP3 inflammasome through Mitofusin-2. (3) Lysosome rupture can release cathepsin B, which in turn activates the NLRP3 inflammasome. Some viruses, such as IAV (Influenza A Virus) NA and AdV (Adenovirus), are known to be associated with this process. (4) Activation of the NLRP3 inflammasome by affecting inflammasome assembly: examples include EV71 3D, ECHO 11, SARS-CoV N, and Zika Virus NS5. (5) Inhibition of NLRP3 inflammasome activation through disruption of assembly and activation: Some viruses, such as MeV, Nipah Virus (Nipah V), SeV, IAV NS1/PB1-F2, EBV miR-BART15, AdV VA RNAI, and ZIKV NS3, inhibit NLRP3 inflammasome activation by interfering with its assembly and subsequent activation. (6) Induction of NLRP3 inflammasome degradation and inhibition of activation: certain viruses, like EV71 2A/3C and HPIV3 C, trigger the degradation of the NLRP3 inflammasome, thereby inhibiting its activation.

**Figure 5 biomolecules-13-01454-f005:**
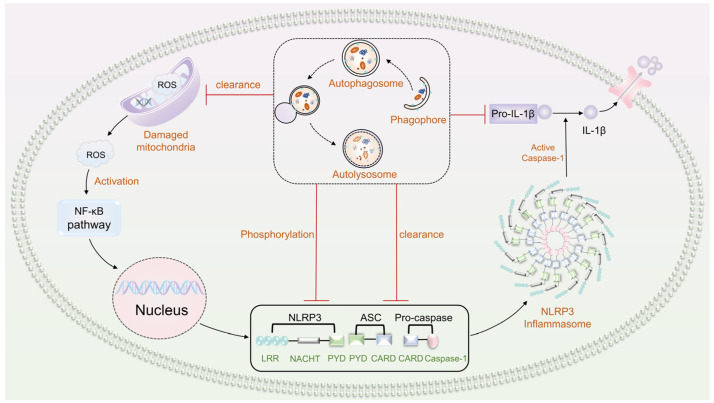
A complex and functionally interrelated regulatory relationship exists between autophagy and the NLRP3 inflammasome. Mitochondrial-derived ROS released from damaged mitochondria can activate the NF-κB pathway, initiating the transcription of NLRP3 and pro-IL-1β. Consequently, this triggers the subsequent activation of the NLRP3 inflammasome. In contrast, autophagy can be induced to eliminate endogenous inflammasomes, such as those originating from damaged mitochondria generating ROS, thus inhibiting the assembly of the NLRP3 inflammasome. Moreover, autophagy can directly target crucial components of the inflammasome, including NLRP3, caspase-1, and ASC, and can also phosphorylate NLRP3. Through the sequestration and degradation of these molecules, autophagy assumes a pivotal role in curbing the assembly and activation of the NLRP3 inflammasome. Additionally, autophagosomes possess the capability to sequester pro-IL-1β and caspase-1, leading to the suppression of IL-1β secretion, thus exerting supplementary regulatory control over the inflammatory process.

**Table 1 biomolecules-13-01454-t001:** Virus-induced autophagy.

Virus	Viral Protein	Mechanism of Action	Reference
(i) Induction of autophagy through interaction with autophagy-related proteins			
EV71	3D	Interacts with Beclin1 to promote EV71 replication in cells.	[26]
HBV	X	Induces Beclin1 ubiquitination and disrupts the Beclin1/BCL2 interaction, thereby promoting autophagy.	[22]
HCV	P7	Binds to Beclin1, inducing autophagy.	[25]
	NS5A	Upregulates BECN1 to induce autophagy.	[24]
	NS5B	Interacts with ATG12 and ATG5 to induce autophagy.	[32]
	NS4B	Associates with Rab34 and Vps5 to initiate autophagy.	[31]
RSV	NS2	Interacts with Beclin1, preventing Beclin1 degradation and enhancing cellular autophagy activity.	[23]
HIV	Gag	Stimulates the formation of autophagosomes by directly interacting with LC3.	[27]
HPIV3	M	Triggers mitochondrial autophagy through interaction with LC3 proteins involved in autophagosome formation.	[28]
FMDV	VP1	Co-localizes with LC3 to form LC3 puncta, inducing autophagy.	[29]
SeV	-	Promotes the conversion of LC3-I to LC3-II. Simultaneously, the expression of CCDC50 significantly increases, and it can interact with p62 and LC3B to induce autophagy.	[30]
VSV			
(ii) Modulation of autophagy signaling pathways for inducing autophagy			
SVV	VP1	Stimulates AKT and AMPK phosphorylation while suppressing mTOR phosphorylation, initiating autophagy through the AKT-AMPK-mTOR pathway.	[33]
SVV	VP3	Activates autophagy through the ERK-mTOR and p38 MAPK-mTOR pathways.	[33]
	3C		
EBV	Rta	Activates autophagy through the ERK signaling pathway.	[40]
CVB3	-	Regulates autophagy by modulating the AMPK/MEK/ERK and Ras/Raf/MEK/ERK signaling pathways.	[39]
CA16	-	Stimulates autophagy through the suppression of the Akt/mTOR pathway and the activation of the MEK/ERK pathway.	[36]
FMDV	VP2	Induces autophagy through the EIF2S1-ATF4-AKT-MTOR cascade.	[37]
EV71	-	Decreases p-mTOR and p-p70S6K expression, contributing to EV71-induced autophagy.	[38]
RV	NSP4	Suppresses host cell IGF1R expression, blocking the downstream PI3K/Akt/mTOR signaling pathway and activating cellular autophagy.	[35]
ZIKV	NS4A	Suppresses the Akt-mTOR signaling pathway, leading to impaired neurogenesis and upregulated autophagy after infecting human embryonic neural stem cells, promoting viral replication.	[34]
	NS4B		
(iii) Induction of autophagy through ER stress			
HCV	-	Induces ER stress, activating PERK, ATF6, IRE1a, CHOP, etc., and further activating or inducing downstream effector factor expression, mediating autophagy occurrence.	[42]
FMDV	-	Induces ER stress and the UPR, subsequently triggering autophagy.	[45]
NDV	NP	Alters ER homeostasis and activates the P-ERK and ATF6 pathways, resulting in ER stress and autophagy occurrence.	[43]
	P		
EMCV	2C	Modulating the expression of proteins related to the UPR pathway through ER stress leads to the induction of autophagy.	[44]
	3D		

**Table 2 biomolecules-13-01454-t002:** Antiviral mechanisms of virus escape mediated by autophagy.

Virus	Viral Protein	Mechanism of Action	Reference
(i) Viral inhibition of autophagosome formation			
KSHV	ORF16	Inhibition of autophagy occurs through interaction with Beclin1.	[51,52]
MHV-68	M11		
γ-herpesviruses	Bcl-2		
KSHV	vFLIP	Inhibiting the binding of Atg3 and subsequent LC3 processing leads to the suppression of autophagy.	[53]
MCV			
HVS			
HSV-1	ICP34.5	Directly targets Beclin1 to block autophagosome formation, enhancing the virus’s neurovirulence.	[54]
HSV-1	Us3	Suppresses autophagy by antagonizing ULK1 and Beclin1. Us3 continuously activates mTORC1 to reinforce ULK1 inhibition and effectively mimics Akt by directly phosphorylating Beclin1.	[55]
FMDV	3C	Inhibits cell autophagy and host antiviral response by cleaving the ATG5-ATG12 complex, reducing LC3II and ATG5-ATG12 expression levels, and promoting viral replication.	[56]
(ii) Viruses block autophagosome–lysosome fusion			
KSHV	-	Blocks autophagosome–lysosome fusion by downregulating Ras-related protein (RAB) 7.	[57]
EBV	-	Reduces autophagosome–lysosome fusion in a Rab7-dependent manner.	[58]
HPIV3	P	Inhibits autophagosome–lysosome fusion by binding to SNAP29 and obstructing its interaction with syntaxin17.	[59]
CVB3	3C	Inhibits autophagy by cleaving SNAP29 and PLEKHM1, disrupting their function in autophagosome–lysosome fusion.	[60]
FMDV	2C	Prevents lysosome–autophagosome fusion by interacting with Beclin1, promoting viral survival.	[61]
HIV-1	Nef	Binding to Beclin1 triggers mTOR activation, TFEB phosphorylation, and its subsequent cytosolic sequestration, culminating in the suppression of autolysosome maturation.	[62,63]
CA16	2C	Blocks autophagosome–lysosome fusion.	[36]
	3C		
IAV	M2	Impairs autophagosome–lysosome fusion and induces the accumulation of autophagosomes, facilitating viral replication.	[64]
SARS-CoV-2	ORF3a	ORF3a disrupts RAB7-HOPS fusion machinery assembly, inhibiting autophagosome–lysosome fusion.	[65]

**Table 3 biomolecules-13-01454-t003:** Viruses promote the activation of the NLRP3 inflammasome.

Virus	Viral Protein	Mechanism of Action	Reference
(i) Ion concentration imbalance mediates the activation of the NLRP3 inflammasome			
HIV	gp120	Promotes NLRP3 inflammasome activation through the CXCR4-Kv1.3 pathway, leading to K^+^ efflux.	[87]
IAV	M2	H37G mutation in the M2 protein enhances NLRP3 activation, possibly due to K^+^ efflux through Na^+^/K^+^ exchange and H^+^ efflux from the Golgi apparatus, triggering NLRP3 inflammasome activation.	[88,89]
MAYV	-	Triggers NLRP3 inflammasome activation through induction of ROS and K^+^ efflux.	[90]
SARS-CoV	ORF3a	Disrupts intracellular ion homeostasis and causes mitochondrial damage, leading to K^+^ efflux and ROS production, resulting in NLRP3 inflammasome activation.	[91]
	E	Forms Ca^2+^ channels in the ER-Golgi intermediate compartment or Golgi membrane, promoting NLRP3 inflammasome activation and excessive IL-1β production.	[92]
FMDV	2B	Activation of NLRP3 signaling 2 is linked to the efflux of K^+^ and the influx of Ca^2+^, both occurring irrespective of ROS and lysosomal protease B involvement.	[93,94]
VSV	-	Induction of cell death and potassium (K^+^) efflux can promote NLRP3 inflammasome activation, leading to the release of IL-1β and cellular pyroptosis and necrosis.	[95]
EMCV	2B	Localizes to the ER and Golgi apparatus, reducing intracellular Ca^2+^ levels, inducing NLRP3 redistribution to the perinuclear region, leading to NLRP3 inflammasome activation.	[96]
HRV	2B	Localizes to the ER and Golgi apparatus, reducing intracellular Ca^2+^ levels, inducing NLRP3 redistribution to the perinuclear region, subsequently triggering NLRP3 inflammasome activation.	[97]
Picornavirus	2B	Reduces Ca^2+^ levels in the ER and Golgi apparatus, resulting in Ca^2+^ influx into the cytoplasm and disrupting ion homeostasis, leading to NLRP3 inflammasome activation.	[98]
RSV	SH	Aggregates in Golgi lipid raft structures, forming selective ion channels that induce NLRP3 translocation from the cytoplasm to the Golgi apparatus, triggering inflammasome activation.	[99]
(ii) Mitochondrial damage activates the NLRP3 inflammasome			
HBV	X	Localizes to mitochondria and enhances NLRP3 inflammasome-mediated inflammation and pyroptosis by upregulating mtROS production.	[100]
SARS-CoV	3a	Disrupts intracellular ion homeostasis and causes mitochondrial damage, resulting in K^+^ efflux and ROS production, ultimately driving NLRP3 inflammasome activation.	[91]
DENV	NS2A	Promotes NLRP3 inflammasome complex assembly and activates caspase-1 and IL-1β secretion via calcium mobilization or disruption of mitochondrial membrane potential. This process also induces the production of ROS.	[101]
	NS2B		
VSV	-	Promotes RIP1-RIP3 interaction, activates DRP1, and facilitates its translocation to mitochondria, causing abnormal fission and damage, leading to NLRP3 inflammasome activation.	[102]
IAV	PB1-F2	Reduced mitochondrial membrane potential leads to accelerated mitochondrial fragmentation and activation of the NLRP3 inflammasome.	[103]
EMCV	-	Induces mitochondrial membrane potential, under which NLRP3 translocates to mitochondria and binds with Mitofusin-2, promoting NLRP3 inflammasome activation.	[104]
(iii) Lysosomal rupture activates the NLRP3 inflammasome			
IAV	NA	Possibly interacts with lysosome-associated membrane proteins 1 and 2, promoting deglycosylation of LAMPs, inducing lysosome rupture, and releasing cathepsins into the cytoplasm, increasing viral load and enhancing cell death.	[105]
AdV	-	Induces lysosomal membrane rupture and cathepsin B release, leading to NLRP3 inflammasome activation.	[106]
(iv) Affecting inflammasome assembly and activating the NLRP3 inflammasome			
SARS-CoV-2	N	Facilitates the binding of NLRP3 to ASC and enhances the assembly of the NLRP3 inflammasome.	[107]
EV71	3D	Directly interacts with NLRP3, forming a 3D-NLRP3-ASC ring structure, enhancing inflammasome assembly and activation, promoting IL-1β production.	[108]
ZIKV	NS5	Forms an NS5-NLRP3-ASC ring structure by binding to NLRP3’s LRR and NACHT domains in the cytoplasm, promoting NLRP3 inflammasome complex assembly.	[109]
ECHO 11	-	Interaction with NLRP3 activates the NLRP3 inflammasome and promotes the assembly of the inflammasome complex.	[110]

**Table 4 biomolecules-13-01454-t004:** Viral Inhibition of NLRP3 Inflammasome Activation.

Virus	Viral Protein	Mechanism of Action	Reference
MV	V	Direct or indirect interaction with NLRP3 to inhibit inflammasome assembly.	[113,114]
SeV	V	Engages with NLRP3 to hinder ASC recruitment, impede ASC oligomerization, and suppress inflammasome activation.	[115]
IAV	NS1	Targets two signals of the NLRP3 inflammasome, NF-κβ in Signal 1 and NLRP3 in Signal 2, inhibiting NLRP3 inflammasome activation.	[116]
IAV	PB1-F2	Binds to the PYD and LRR domains of NLRP3, maintaining NLRP3 in a self-inhibited state, thereby preventing the NEK7-NLRP3 interaction and inhibiting inflammasome assembly. This, in turn, reduces pyroptosis in infected cells and promotes viral evasion.	[117]
EBV	mir–BART15	Targeting miR-223 inhibits the translational expression of NLRP3.	[118]
AdV	VA RNAI	Disrupts the interaction between protein kinase R (PKR) and ASC, hinders the oligomerization of ASC protein, and suppresses the activation of the NLRP3 inflammasome.	[119]
ZIKV	NS3	Reduces caspase-1 and IL-1β secretion.	[120]
EV71	2A	By cleaving the NLRP3 protein at the Q225-G226 or G493-L494 junctions using 3C and 2A proteases, the activation of the inflammasome is inhibited, thereby evading immune surveillance.	[121]
	3C		
HPIV3	C	Engages with the NLRP3 protein and enhances its proteasomal degradation, thereby obstructing inflammasome activation.	[122]

## Data Availability

Not applicable.

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
