# Peer review of "Autophagy and Inflammation: Regulatory Roles in Viral Infections"

_biomolecules, 2023, doi:10.3390/biom13101454_

Round 1

Reviewer 1 Report

Li Chen et al. have provided a timely and authoritative review paper on "Autophagy and Inflammation: Regulatory roles in viral infections", in which the roles of the three elements involved, autophagy, viruses and inflammation are clearly differentiated and discussed. The tables and figures are clear and useful.Overall, it is a good review.

Author Response

Thanks for your positive feedback on this manuscript.

Reviewer 2 Report

The manuscript entitled "Autophagy and Inflammation: Regulatory roles in viral infections" Title, abstract and overall rationale of this review article is written well and highly satisfactory. Still, there are some minor concerns, which needs to be addressed before publication.

1) Abstract and keywords parts written well.

2)  The necessity and innovation of the article should be presented to the introduction.

3) Author wrote about the inflammation happen during virus activation. These all section is written well and describe details. However, I suggest author to add one paragraph about the Th1 and Th2 response during these inflammation and what is the role of these cytokines.

4) In conclusion section author must be write the limitation of this study and significance. Moreover, author also need to write future prospective of this study.

5) Author need minor correction of grammatical mistake and typo error in the manuscript.  

6) Some references are too long and author need to revise for example reference no 57, 72 and other. I suggest author to revise if other latest manuscript is available in the same information.

English quality is good.

Author Response

The manuscript entitled "Autophagy and Inflammation: Regulatory roles in viral infections" Title, abstract and overall rationale of this review article is written well and highly satisfactory. Still, there are some minor concerns, which needs to be addressed before publication.

  • Abstract and keywords parts written well.

Thanks!

  • The necessity and innovation of the article should be presented to the introduction.

Thank you for your advice. We have added the necessity and innovation of this article at the end of the introduction and marked it in blue.

  • Author wrote about the inflammation happen during virus activation. These all section is written well and describe details. However, I suggest author to add one paragraph about the Th1 and Th2 response during these inflammation and what is the role of these cytokines.

Thank you for your invaluable feedback. We greatly appreciate your comments and fully endorse your suggestion. Consequently, we have integrated the requested addition into the manuscript. In response to your recommendation, we have introduced a new paragraph within the section that delves into inflammation and virus activation. This paragraph is aptly titled 'The Role of Cytokines in the Antiviral Immune Process' and furnishes an overview of Th1 and Th2 responses during inflammation, elucidating the pivotal role of cytokines in these mechanisms. You can locate this freshly incorporated paragraph on page 16.

  • In conclusion section author must be write the limitation of this study and significance. Moreover, author also need to write future prospective of this study.

Thank you for your comments. We have revised the conclusion section to include discussions on this study's limitations, significance, and future prospects.

  • Author need minor correction of grammatical mistake and typo error in the manuscript.  

We have reviewed the entire manuscript, correcting grammar and typo errors.

  • Some references are too long and author need to revise for example reference no 57, 72 and other. I suggest author to revise if other latest manuscript is available in the same information.

Thanks for the suggestion. We have revised the references to cite the most recent literature available.

Reviewer 3 Report

Chen et. al in this extensive review on autophagy - the process and during virus infection and inflammation through the NLRP3-inflammasome axis describe the individual aspects and then discuss the crosstalk between these two phenomena during viral infections. The highlight of this review is the detailed summary figures and tables that make a great referencing tool.

I recommend that this manuscript can be accepted with minor revisions which are as follows:

1. Rephrase in abstract lines 16 and 17, appears confusing to read inflammation regulates inflammatory responses. Suggested word choice is 'negative feedback loop.'

2. Table 3 - SARS-CoV is written incorrectly at Sars cov and for each section it would read better if the SARS viral proteins are written together 

3. Table 4 - Measles in misspelled

4. Sendai virus included for inflammation but left out in the autophagy section

Author Response

Chen et. al in this extensive review on autophagy - the process and during virus infection and inflammation through the NLRP3-inflammasome axis describe the individual aspects and then discuss the crosstalk between these two phenomena during viral infections. The highlight of this review is the detailed summary figures and tables that make a great referencing tool.

I recommend that this manuscript can be accepted with minor revisions which are as follows:

  1. Rephrase in abstract lines 16 and 17, appears confusing to read inflammation regulates inflammatory responses. Suggested word choice is 'negative feedback loop.'

Thanks for your suggestion. We have modified this sentence to “In turn, inflammation can establish negative feedback loops by modulating autophagy to suppress excessive inflammatory reactions.”

  1. Table 3 - SARS-CoV is written incorrectly at Sars cov and for each section it would read better if the SARS viral proteins are written together

Thank you for your suggestion. We have revised Table 3 by grouping the SARS viral proteins.

  1. Table 4 - Measles in misspelled

Corrected.

  1. Sendai virus included for inflammation but left out in the autophagy section

Thanks for your comments. To maintain consistency throughout the manuscript, we have implemented the following adjustments:

  1. Descriptions of Sendai virus (Sev) and vesicular stomatitis virus (VSV) have been integrated into the "Autophagy and Viruses" section, as outlined on page 4 of the manuscript (lines 164-168).
  2. Sev and VSV have been included in Table 1 to ensure comprehensive and clear representation.